# Neoadjuvant PARP inhibitor scheduling in BRCA1 and BRCA2 related breast cancer: PARTNER, a randomized phase II/III trial

Poly (ADP-ribose) polymerase inhibitors (PARPi) exploit DNA repair deficiency in germline BRCA1 and BRCA2 pathogenic variant (gBRCAm) cancers. Haematological toxicity limits chemotherapy-PARPi treatment combinations. In preclinical models we identified a schedule combining olaparib and carboplatin that avoids enhanced toxicity but maintains anti-tumour activity. We investigated this schedule in a neoadjuvant, phase II-III, randomised controlled trial for gBRCAm breast cancers (ClinicalTrials.gov ID:NCT03150576; PARTNER). The research arm included carboplatin (Area Under the Curve 5, 3-weekly); paclitaxel (80 mg/m², weekly) day 1, plus olaparib (150 mg twice daily) day 3-14 (4 cycles), followed by anthracycline-containing chemotherapy (3 cycles); control arm gave chemotherapy alone. The primary endpoint, pathological complete response rate, showed no statistical difference between research 64.1% (25/39); control 69.8% (30/43) ($p = 0.59$). However, estimated survival outcomes at 36-months demonstrated improved event-free survival: research 96.4%, control 80.1% ($p = 0.04$); overall survival: research 100%, control 88.2% ($p = 0.04$) and breast cancer specific survival: research 100%, control 88.2% ($p = 0.04$). There were no statistical differences in relapse-free survival and distant disease-free survival, both were: research 96.4%, control 87.9% ($p = 0.20$). Similarly, local recurrence-free survival and time to second cancer were both: research 96.4%, control 87.8% ($p = 0.20$). The PARTNER trial identified a safe, tolerable schedule combining neoadjuvant chemotherapy with olaparib. This combination demonstrated schedule-dependent overall survival benefit in early-stage gBRCAm breast cancer. This result needs confirmation in larger trials.

## PARP inhibitors in neoadjuvant gBRCAm

The discovery that small-molecule inhibitors of PARP as single agents could selectively kill BRCA1 and BRCA2 deficient cancer cells[1,2] led to new therapeutic approaches for patients with gBRCAm and other associated DNA damage response (DDR) gene aberrations, including deficiencies in the homologous recombination repair (HRR) pathway[3,4]. PARPi monotherapy is currently approved for the treatment of gBRCAm-related cancers: breast, ovary, pancreas and prostate[5]. Greater efficacy of PARP inhibition has been observed in earlier lines of therapy, with olaparib trials for both breast cancer (OlympiA vs OlympiAD trials[6,7]) and ovarian cancer (SOLO1 vs SOLO2 trials[8,9]). There is an increased prevalence of gBRCAm breast cancer within the triple negative breast cancer (TNBC) sub-type[10,11]. The backbone of neoadjuvant chemotherapy for TNBC and

✉ e-mail: ja344@cam.ac.uk

gBRCAm patients currently includes carboplatin with taxanes and anthracyclines[12]. In neoadjuvant chemotherapy given before surgery, the pathological complete response (pCR) endpoint, defined as no invasive cancer at surgery in the breast and axilla, has been shown in TNBC to be associated with improved survival outcomes[13,14]. However, two trials have shown that this association is less certain for tumours in gBRCA patients[15,16]. At the time the PARTNER trial was designed (2014–2015), neither immunotherapy nor carboplatin was routinely used for TNBC or gBRCAm breast cancers.

PARPi monotherapy in neoadjuvant gBRCAm TNBC has shown pCR rates of 40% for niraparib[17] and 45.8% for talazoparib[18], compared to 65–67% for combination chemotherapy[19]. Both platinum agents and PARPis, such as olaparib, generate DNA double-strand breaks (DSBs), mostly during the DNA synthesis phase (S-phase) of the cell cycle[20], but platinum agents induce greater levels of DNA damage than PARPis. While the combination of carboplatin and PARPi will induce greater DNA damage and provide more anti-cancer activity than either agent alone[21], delivering the concurrent combination is hampered by overlapping bone marrow (BM) toxicity[22].

Here we show the preclinical experiments that identified the gap scheduling and how the incorporation of the gap scheduling into a neoadjuvant randomised controlled clinical trial affects relapse-free and overall survival outcomes.

## Results

### Results of the preclinical schedule optimisation experiments
Between 2012 and 2014, prior to opening the PARTNER trial to recruitment in 2016, we undertook a series of preclinical experiments to identify the optimal scheduling strategy for PARPi and carboplatin combination treatment, which minimised BM toxicity, but still enhanced tumour volume reduction (efficacy) compared to either agent alone. Two separate animal model systems were used. BM toxicity assessments required an animal with a fully functional immune system, whereas the patient-derived tumour explant (PDX) in vivo model work, assessing tumour volume reduction, required a nude (immune-suppressed) animal model. Mouse BM DDR is different from that of humans[21,23], as a result, these models underestimate the extent of BM toxicity in comparison to that seen in humans. However, rat BM DDR is more similar to human DDR[23]. We, therefore, used a rat BM model[24] to analyse DNA damage and BM toxicity for carboplatin-based chemotherapy, with and without the addition of olaparib.

To identify an optimal PDX model to study platinum and olaparib DNA damage induction and the anti-tumour efficacy of the combination, we needed a PDX model that was neither too sensitive nor too resistant to either single agent for the combination effect to be differentially assessed. Three TNBC PDX models, with previously defined and different levels of platinum sensitivity, were assessed at day 28 of treatment for their response to olaparib alone and with a concurrent platinum combination (Methods and Supplementary Fig. 1). BRCA2m PDX HBCx-17 demonstrated the greatest combination benefit over olaparib and platinum single agent treatments, providing the best opportunity to assess the effect of combination scheduling in preclinical optimisation studies.

To identify optimal PARPi/carboplatin combination dosing schedules, we first assessed the DNA damage induction and repair kinetics of carboplatin using immunohistochemistry (IHC) and an antibody detecting phosphorylated histone H2AX (γH2AX), a biomarker of DNA DSBs[25]. Figure 1A provides representative examples of the γH2AX IHC staining in rat BM sections with vehicle control at 6 h and 72 h after 50 mg/kg carboplatin treatment. Also shown are the vehicle control and 72 h post-carboplatin samples of PDX HBCx-17 tumour sections. The 50 mg/kg dose of carboplatin is approximately equivalent to a human carboplatin dose of area under the curve (AUC) 6. Figure 1B, C shows the quantification of γH2AX IHC in rat BM and human PDX tumour, respectively. The data demonstrate that DNA damage in rat BM peaks at

6 h post-treatment and is virtually undetectable after 48 h (Fig. 1B). In contrast, the DNA damage detected in the BRCA2m PDX HBCx-17 tumour tissue continued to increase beyond 48 h (Fig. 1C). This is consistent with higher levels of endogenous DNA damage and loss of DDR capability that represents a hallmark of cancers compared to normal tissue[20]. These data suggest that at 48 h the carboplatin induced DNA damage in BM cells, if resolved, will not be further potentiated by the addition of a PARPi. However, in the PDX tumour tissue, PARPi treatment 48 h after carboplatin may still enhance the DNA damage effect, where carboplatin-induced DNA damage is still ongoing.

### Selection of a gap schedule to limit bone marrow toxicity while maintaining efficacy
Using flow cytometry to quantify rat BM cells on day 7 following treatment initiation ('Methods'), we assessed the effect of introducing a 24, 48, 72 and 96 h gap between the carboplatin and olaparib treatments (Fig. 2A). A 24 h gap did not change the depth of the day 7 post-carboplatin treatment nadir in CD90+ BM cells seen with the concurrent olaparib combination, whilst 48, 72 and 96 h gaps before the olaparib treatment all alleviated the enhanced combination toxicity effect on BM cells. This suggested that a gap schedule, such as a 48 h gap, had the potential to reduce the combination toxicity seen with concurrent PARPi and platinum-based chemotherapy.

As concurrent treatment was expected to have the greatest efficacy, but was associated with prohibitive BM toxicity clinically, we used the shortest gap that provided better BM tolerability for the combination in our in vivo model HBCx-17, namely a 48 h gap schedule. To be certain that introducing the 48 h gap would not result in a loss of anti-tumour benefit of the olaparib combination, we assessed efficacy in HBCx-17 comparing the 48 h gap schedule to the single-agent treatments of clinically relevant olaparib and carboplatin doses. The 48 h gap schedule combination maintained greater anti-tumour activity compared to the single agents alone (Fig. 2B).

These preclinical experimental results therefore provided a rationale for considering the inclusion of a 48 h gap schedule of carboplatin and olaparib into the PARTNER trial design.

### Design of the gBRCAm PARTNER trial
PARTNER is a prospective, phase II-III, randomised controlled clinical trial, with a multi-arm multi-stage pick the winner design[26], which recruited patients with gBRCAm. Stage 1 examined the safety of combining olaparib with carboplatin and paclitaxel. Stages 1 and 2 compared two different schedules of olaparib and carboplatin with paclitaxel to identify optimal olaparib scheduling (randomisation 1:1:1). Stage 3 evaluated the selected olaparib schedule compared with chemotherapy alone (randomisation 1:1). The primary endpoint was pCR, and secondary endpoints included event-free survival (EFS) and overall survival (OS). Patients in the control arm received chemotherapy alone. Chemotherapy was administered on day 1; carboplatin AUC5 intravenously (i.v.) with paclitaxel 80 mg/m² i.v. on day 1, 8 and 15 every 3 weeks for four cycles, followed by three cycles of standard anthracycline-based chemotherapy before surgery. During stages 1 and 2 there were two randomised investigational arms which contained olaparib. In the first investigational arm, established from the preclinical studies, the schedule was olaparib from day +3 to day +14, designated the gap schedule (research) arm. The schedule in the second investigational arm was olaparib from day −2 to day +10, designated the non-gap schedule (dropped) arm. Both schedules used olaparib 150 mg tablets twice daily for 12 days, given with each of the four cycles of carboplatin-paclitaxel regimen only (Fig. 3A). The rationale for adding the second investigational arm with olaparib starting 2 days before chemotherapy was due to concerns that carboplatin-induced nausea and vomiting might limit the patient's ability to take olaparib tablets once chemotherapy commenced.

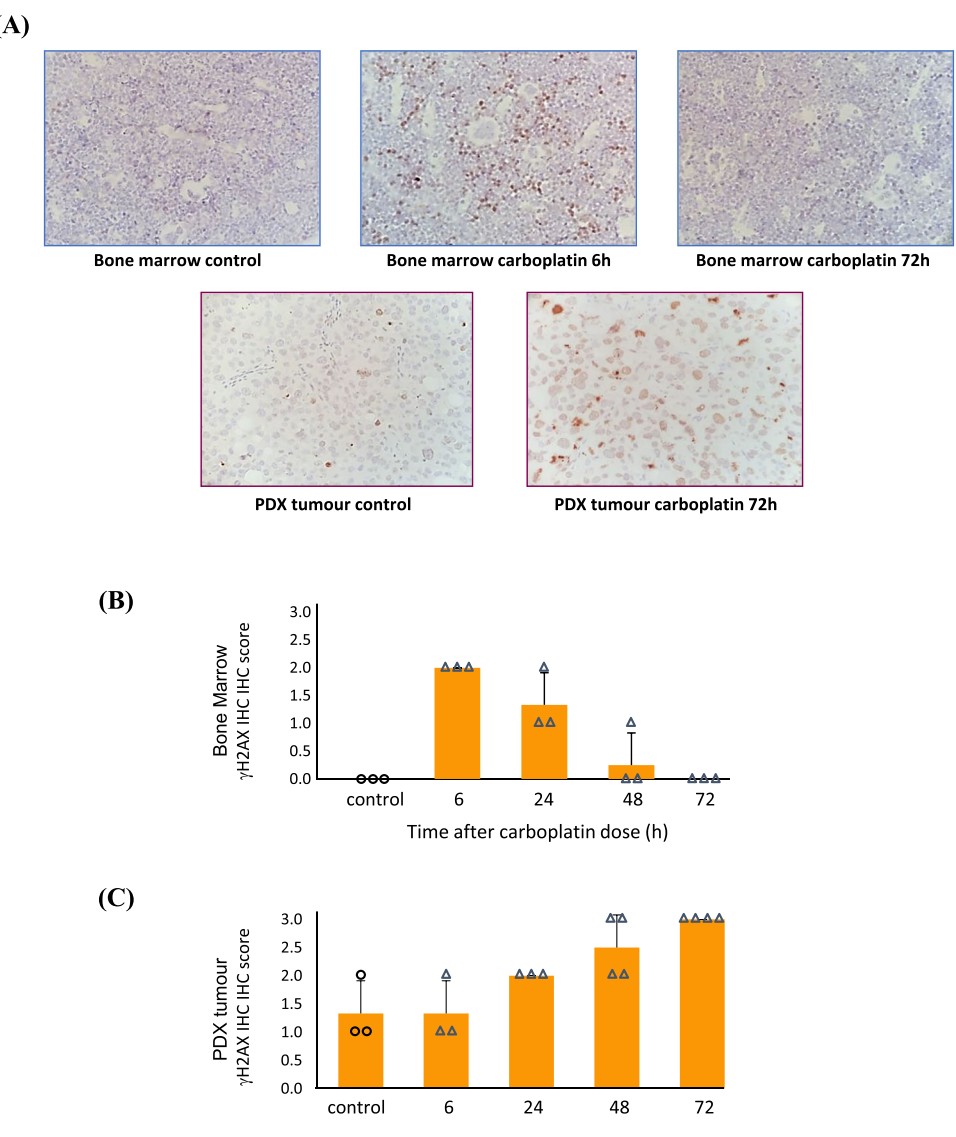

**Fig. 1 | Comparison of DNA damage repair kinetics in bone marrow versus tumour cells. A** Examples of γH2AX immunohistochemistry (IHC) staining (brown colour) of sections of rat bone marrow (top panel) with vehicle control treatment or carboplatin (50 mg/kg) at 6 h and 72 h post treatment. The lower panel is a representative image of PDX tumour γH2AX staining. Quantification of γH2AX from both rat bone marrow (**B**) and PDX tumour (**C**) is shown using the pathologists' scoring system of 0–3 from three independent biological replicates (or four independent biological replicates in the case of the PDX 48 h and 72 h data). Individual scores for vehicle control (circles) or platinum treatment (triangles) for the biological replicates are also shown, as is the group mean (orange bars with SD error bars). Source data is provided as a Source Data file. γH2AX staining is indicative of DNA damage induction and repair over time and shows that carboplatin treatment is resolved in rat bone marrow after 48 h, while in the PDX tumour, DNA damage is still increasing at 72 h post treatment.

Stage 1 identified no major safety issues. At completion of stage 2, the independent data monitoring and safety committee (IDMSC) was asked to pick the winner between the olaparib-containing investigational arms based on pre-specified criteria, which included safety, efficacy and patient compliance/convenience. The IDMSC, in their advisory role, recommended that we continue with the gap schedule research arm (Supplementary Note 1). A protocol-defined, pre-planned interim analysis was conducted after approximately 50% of the gBRCAm participants had pCR data available. Results from this analysis were reviewed by the IDMSC, which concluded that, while the pre-determined statistical criteria for futility (conditional power < 15%) were not met, it was clear that the primary endpoint (significant improvement in pCR) was unlikely to be achieved. Moreover, secondary endpoint results such as EFS and OS were, in future patients, likely to be confounded by the licensed availability

from 10/05/2023 of twelve months of adjuvant olaparib for gBRCAm patients with non-pCR, as per the OlympiA trial[6]. The IDMSC took all these factors into account and advised stopping recruitment and unblinding the results to the study team. This advice was considered by the trial steering committee and trial management group, who then stopped the trial. The numbers of gBRCAm participants randomised and analysed at each stage are shown in Supplementary Table 1.

## Results of the gBRCAm PARTNER trial
**Patients and treatment.** From June 2016 to May 2023, 108 gBRCAm patients were randomised (across all stages) from 23 UK centres into the control arm (n = 47); the gap schedule 'research' arm (n = 39), and the non-gap schedule dropped arm (n = 22). Two patients in the control arm opted out of the study after randomisation and did not receive

(A)

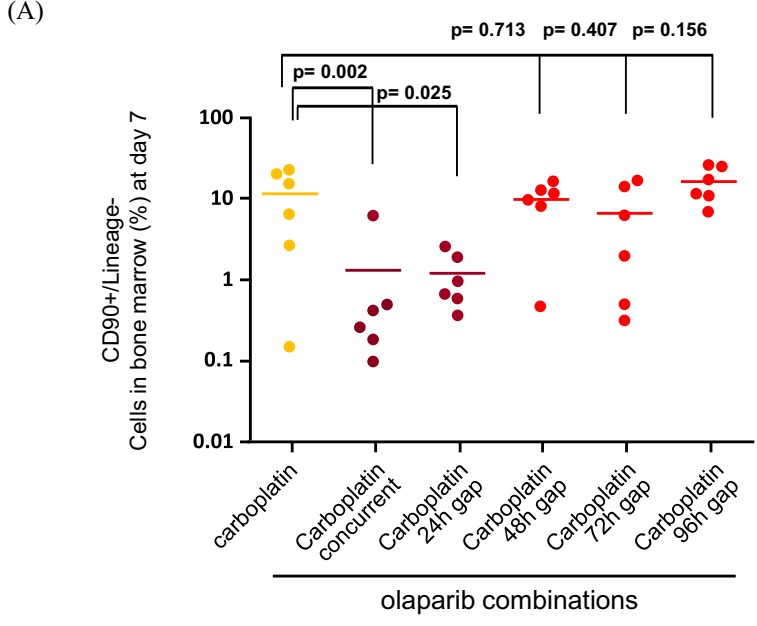

(B)

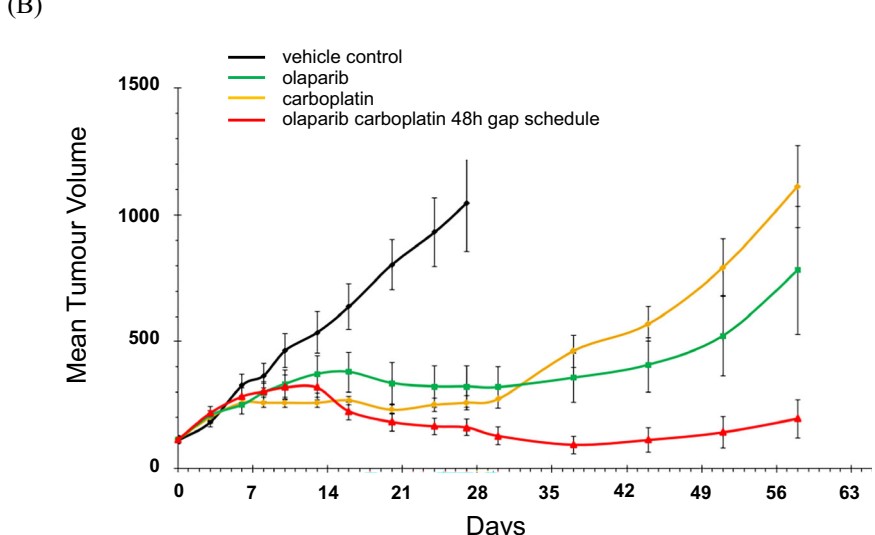

**Fig. 2 | Introducing a 48 h gap between carboplatin and olaparib treatments ameliorates the combination bone marrow toxicity effect while maintaining combination anti-tumour efficacy. A** Flow cytometry quantification of CD90+/Lineage− multipotent progenitor stem cells indicates that a 24 h gap between the carboplatin and olaparib combination is not sufficient to reduce bone marrow toxicity compared to concurrent treatment. Introduction of a 48, 72 or 96 h gap does reduce the combination toxicity effect to that seen for carboplatin alone. ANOVA statistical analysis of CD90+ cell levels from 6 biological replicates used a two-sided Student's *t* test. *P*-values for the statistical significance of comparisons between the % CD90+/Lineage− multipotent progenitor stem cells treated with carboplatin alone and those involving concurrent or gap scheduling combinations with olaparib are indicated. **B** The use of a 48 h gap schedule for the carboplatin/olaparib combination still maintains greater anti-tumour efficacy in the TNBC gBRCAm PDX model HBCx-17 than the effects of either olaparib or carboplatin alone. Treatment represented by the black line is 28 d of vehicle control, the green line is 28 days (28D) daily 100 mg/kg olaparib starting on Day 1 (D1), the yellow line is a single D1 dose of 50 mg/kg carboplatin and the red line the single D1 50 mg/kg carboplatin and 28D daily 100 mg/kg olaparib treatment starting on D3. Mean tumour volumes are plotted along with error bars shown with ±SEM. Statistical significance was evaluated using a one-tailed *t*-test from 10 independent biological replicates for the vehicle control and 9 biological replicates for the other three treatment arms. Source data is provided as a Source Data file.

any treatment (Fig. 3B). As a result, the modified intention-to-treat (mITT) population consisted of 84 patients (research arm, *n* = 39; control arm, *n* = 45). The data cut-off date was 30th November 2023.

Supplementary Table 2 summarises the demographics and pre-treatment disease characteristics in the control and research arms. In the research arm, 89.7% of patients received at least 80% of the planned olaparib. All patients in both arms received at least 80% of paclitaxel. In the research arm 97.4% received at least 80% of carboplatin compared to 91.1% in the control arm (Supplementary Table 3). Surgery was carried out after the treatment was completed.

## Efficacy

Evaluable outcome data (pCR) were available in 82/84 patients (97.6%; research, *n* = 39; control, *n* = 43) on 30th November 2023. Two participants had missing or unevaluable pCR data (control arm). In the research arm pCR rate was 64.1% (25/39) and 69.8% (30/43) in the control arm, with a difference of −5.7% (95% CI −25.8% to 14.6%, *p*-value = 0.586) (Fig. 4A). No differences in pCR rates were observed in the pre-specified subgroups, including when imputing missing data over a range of plausible assumptions (Supplementary Fig. 2 and Supplementary Table 4). Although the proportion of patients

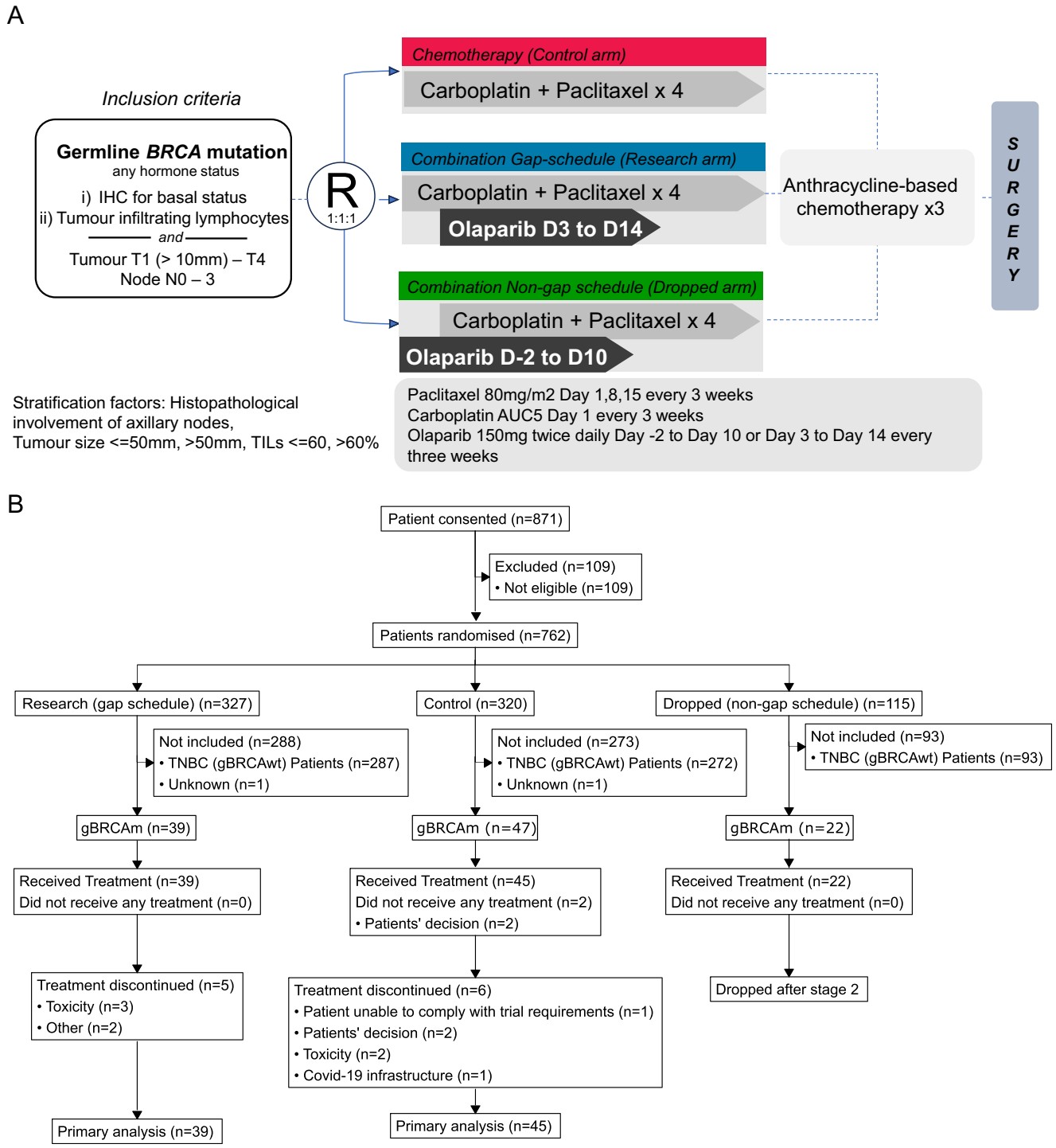

**Fig. 3 | PARTNER Trial gBRCAm cohort schema and CONSORT diagram. A** gBRCAm cohort trial flow chart. **B** Trial Consort Diagram. The first main reason for treatment discontinuation was reported.

with pCR was higher when tumours had tumour infiltrating lymphocyte (TILs) ≥ 60% (76.9%) compared to those with TILs < 60% (62.5%), the difference was not significant (14.4%; 95% CI −8.0% to 33.1%, p value = 0.196; Supplementary Fig. 3). A post hoc analysis, including patients who were gBRCA wild-type (gBRCAwt) TNBC recruited into the PARTNER trial[13] and reported separately, was performed. In this analysis, the pCR rates in the cohort of gBRCAm patients (55/82 [67.1%]) were higher than for those with TNBC (gBRCAwt) (281/543 [51.7%]); difference 15.3% (95% CI 3.8 to 25.5%, p = 0.009) (Fig. 4B).

Pre-planned analyses included survival outcomes as secondary endpoints (Supplementary Note 2). After a median follow-up of 42.0 months, ten patients (1 research; 9 control) had an event. The estimated 36-month EFS was 96.4% (95% CI, 89.8–100%) in the research arm, and 80.1% (95% CI, 68.7–93.5%) in the control arm (log-rank p = 0.04). Six patients died (0 research; 6 control); the estimated 36-month OS was 100% (research arm) and 88.2% (95% CI, 79.1–98.5%; control arm; log-rank p = 0.039) (Fig. 5A, B). Breast cancer-specific survival favours the research arm (p = 0.039). More distant recurrences, local recurrence and second primary cancers were observed in

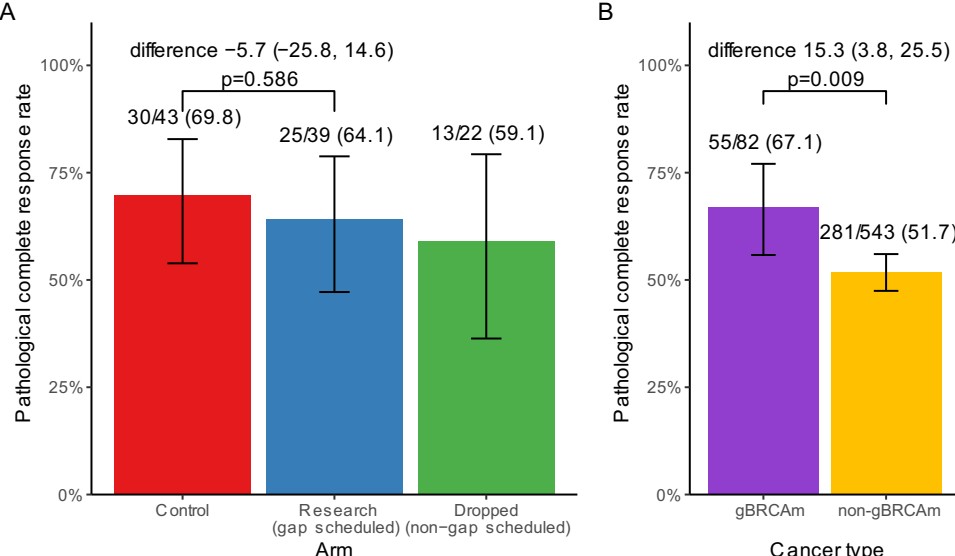

**Fig. 4 | Efficacy endpoints of the gBRCAm cohort in modified intention-to-treat (mITT) patients. A** Pathological complete response (pCR) rate by treatment arm in gBRCAm patients. **B** pCR by cancer subtype in the entire trial population, excluding the dropped arm. The point estimation is the pCR response rate, and the error bar represents the 96% confidence interval of the proportion based on the Clopper–Pearson method. The statistical test was based on the two-sided chi-squared test. Source data is provided as a Source Data file.

the control arm. These differences were not statistically significant based on the log-rank test for local and distant disease-free survival ($p = 0.2$), relapse free survival ($p = 0.2$), and time to second primary cancers ($p = 0.2$)) (Supplementary Fig. 4A–E). A detailed breakdown of the event types, including by individual patient is shown in Supplementary Tables 5A, B. A few imbalances in baseline variables between research and control arms were observed (Supplementary Table 6). The imbalances sometimes favour the control arm (e.g. more patients with high TILs score) and sometimes the research arm (e.g. fewer lymph node positive patients); nonetheless, in all cases, the event rate follows the same trend as for survival. The secondary endpoints of radiological response and residual cancer burden will be reported separately when the data are fully available.

A post hoc analysis, including the non-gap schedule (dropped) arm, indicated that this arm had worse survival outcomes in comparison to both the control and the research arm. Nineteen patients (1/39 research, 9/45 control, 9/22 dropped) had an event. Estimated 36 months EFS was 96.4% (95% CI, 89.8–100%) in the research arm, 80.1% (95% CI, 68.7–93.5%) in the control arm and 66.7% (95% CI, 49.3–90.2%) in the dropped arm (log-rank $p = 0.007$). Twelve patients died (0/39 research; 6/45 control; 6/22 dropped). The estimated 36 months OS was 100% in the research arm, 88.2% (95% CI, 79.1–98.5%) in the control arm, and 72.7% (95% CI, 56.3–93.9%) in the dropped arm (log-rank $p = 0.008$) (Fig. 5C, D).

Figure 5E, F shows the Kaplan–Meier curves of EFS and OS by pathological response and treatment arm. No differences in the EFS (log-rank $p = 0.80$) and OS (log-rank $p = 0.50$) were observed in pCR patients compared to non-pCR patients. The estimated 36 m EFS was 89.1% (95% CI, 80.4–98.6%) in the pCR patients and 86.9% (95% CI, 74.0–100%) in the non-pCR patients. The estimated 36 m OS rate was 96.2% (95% CI, 91.1–100%) in the pCR patients and 92.0% (95% CI, 82.0–100%) in the non-pCR patients. Similarly, no differences in RFS (long-rank $p = 0.20$) were observed. An estimated 36 m RFS was 95.9% (95% CI, 90.5–100%) in the pCR patients and 86.9% (95% CI, 74.0–100%) in the non-pCR patients (Supplementary Fig. 4F).

## Safety
Eighty-four (39 research; 45 control) patients were evaluated for safety. Early treatment cessation occurred in 11 patients (5 research, 6

control). The research arm experienced more grade ≥3 adverse events (AEs) (76.9% (95% CI 60.7–88.9%) versus 60.0% (95% CI 44.3–74.3%)), including thrombocytopenia and non-febrile neutropenia. The serious adverse events (SAEs) related to carboplatin and paclitaxel were higher in the control arm. However, treatment discontinuation rates due to toxicity were similar across treatment arms (7.7% research; 8.9% control) (Supplementary Tables 7–10). The quality of life (QoL) is comparable between the two arms (Supplementary Fig. 5). See also Supplementary Note 3 and full protocol Supplementary Note 4.

## Rationale for differences in outcomes of the olaparib-carboplatin schedules from preclinical gBRCAm model data
The PARTNER trial gBRCAm EFS and OS data clearly demonstrate the significant survival benefit for patients in the gap schedule (research) arm, but also illustrate the critical impact of scheduling on clinical outcome in the gBRCAm subtype. The underlying mechanism for this result needs further investigation. To gain some initial insights into why there was a difference in survival, we carried out a preclinical experiment to assess the effects of the two different schedules in a gBRCA1m TNBC cell line, SUM149PT, which has previously been used in several preclinical studies to assess PARPi activity[27]. Due to the innate sensitivity of SUM149PT cells to olaparib and carboplatin, it was not possible to exactly reproduce the long-term treatments used in the clinical trial schedule in an in vitro cell line experiment. However, we assessed the effects of different combination schedules on SUM149PT cell cycle profiles, DNA damage induction based on γH2AX, as well as the effects of different schedules on cell viability (Fig. 6A–D and Supplementary Fig. 6). These data suggest that differences in the order of olaparib and carboplatin can result in treatment-induced differences in cell cycle profiles, the degree of DNA damage induction and cell viability, with the carboplatin first schedule being more effective.

## Discussion
The PARTNER trial investigated whether the addition of olaparib to neoadjuvant carboplatin-containing chemotherapy, followed by anthracyclines, improved the pCR rate and EFS and OS in patients with early-stage TNBC gBRCAwt and/or gBRCAm breast cancers. The TNBC gBRCAwt and gBRCAm cohorts were each independently powered for

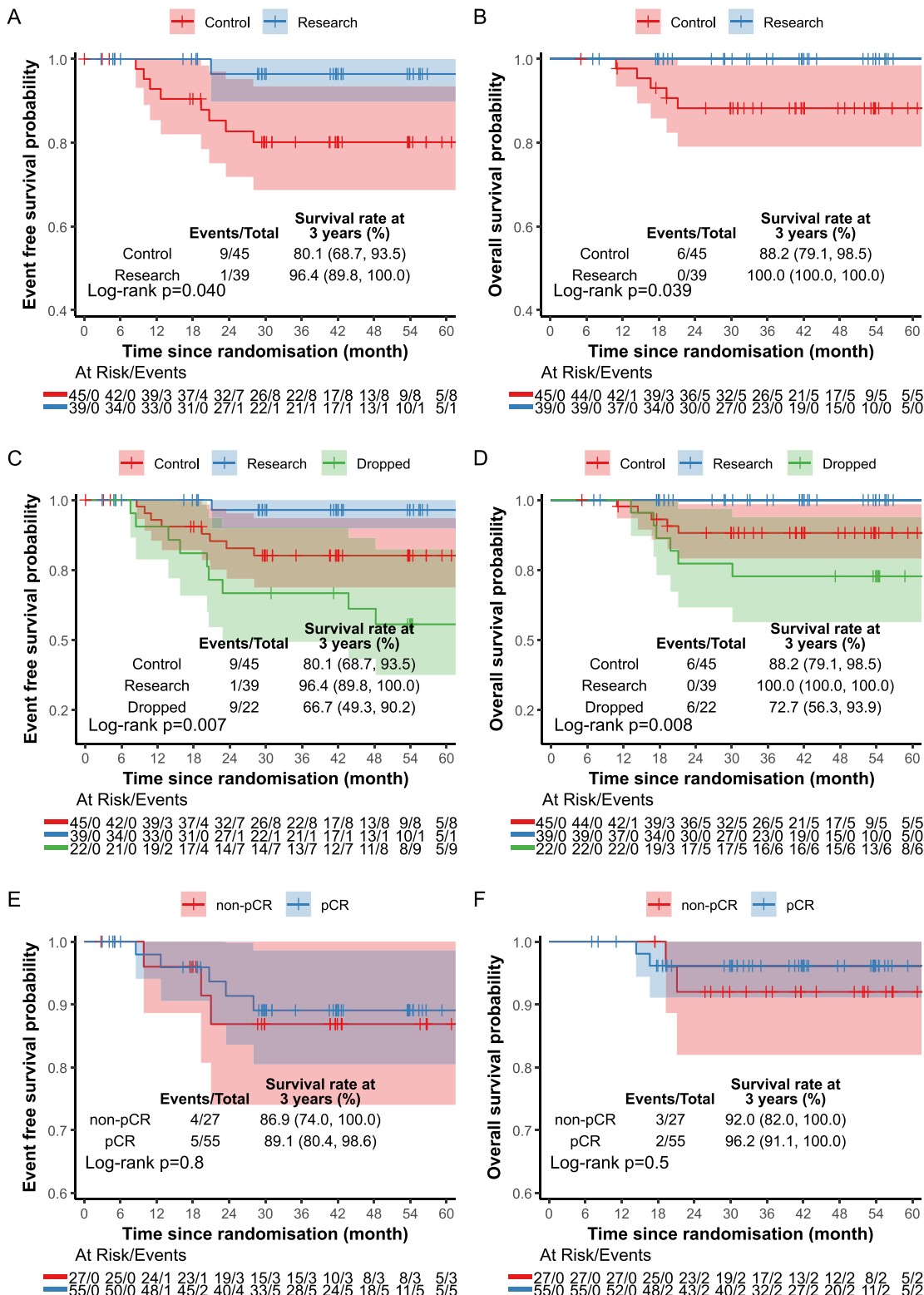

**Fig. 5 | Kaplan–Meier curves of time to event outcomes. A** Event-free survival by treatment arm (control vs research). **B** Overall survival by treatment arm (control vs research). **C** Event-free survival by treatment arm (including the dropped arm). **D** Overall survival by treatment arm (including the dropped arm). **E** Event-free survival by pathological complete response (excluding the dropped arm). **F** Overall survival by pathological complete response (excluding the dropped arm) in the mITT population. No adjustments were made for multiple comparisons. Source data is provided as a Source Data file.

the primary endpoint (pCR) analysis and have been reported separately. The gBRCAm cohort reported here showed markedly different results from the TNBC gBRCAwt cohort[13]. The results of the gBRCAm cohort demonstrate that the gap schedule research arm does not improve pCR rates but significantly improves EFS and OS. The TNBC gBRCAwt[13] cohort, reported elsewhere, showed no benefit from the addition of olaparib in either pCR rates or EFS and OS. In addition, the TNBC (gBRCAwt) cohort did not demonstrate a differential response

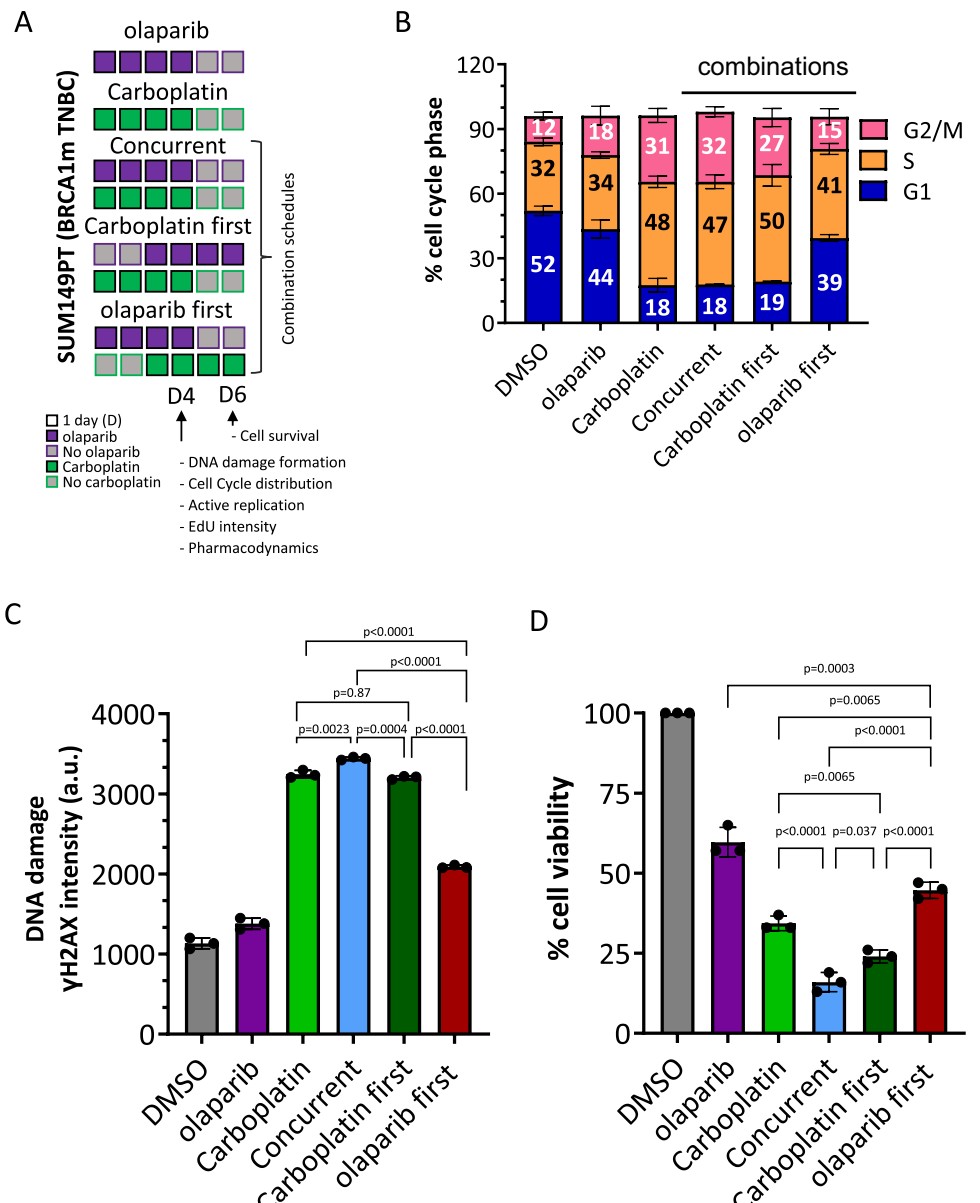

**Fig. 6 | Analysis of different olaparib and carboplatin schedules in gBRCAm SUM149PT cells. A** Schematic of single-agent and combination schedules of olaparib and carboplatin in TNBC gBRCAm SUM149PT cells. Squares represent treatment days with purple for olaparib and green, carboplatin treatment. **B** Cell cycle distribution analysis of SUM149PT. Day 4 (D4) analysis following olaparib (1 μM) and/or carboplatin (10 μM) treatment (mean values ± SD from three independent biological replicates). **C** DNA damage assessment at D4 of SUM149PT cells using γH2AX. Following olaparib (1 μM) and/or carboplatin (10 μM) treatment, γH2AX foci were visualised by immunofluorescence staining (mean values ± SD from three independent biological replicates, a.u. arbitrary units). **D** Cell viability of SUM149PT cells at day six after olaparib (0.3 μM) or carboplatin (1 μM) single agent treatments, concurrent combination or carboplatin first or olaparib first combination schedules. Viability was assessed using a cell titre glow assay (see methods). Shown are the % viability for each treatment (mean values ± SD, from three independent biological replicates). Also provided are the *p*-values from one-way ANOVA multiple comparisons. Source data is provided as a Source Data file.

between the research arm and the dropped arm for pCR rates, EFS or OS outcomes, whereas the gBRCAm did.

The three most important findings from this study are: first, that combination carboplatin-containing chemotherapy plus olaparib in the gap schedule research arm is safe, tolerable and improves EFS and OS in gBRCAm patients; second, that optimal drug scheduling is critical to enable delivery of this treatment and improve outcomes; and third, in this trial there is a lack of association between pCR/non-pCR status and survival outcomes in gBRCAm patients.

The preclinical work that helped design the PARTNER trial research arm demonstrated that introducing a 48 h gap between the carboplatin and olaparib treatments could reduce toxicity, whilst maintaining anti-tumour activity. The important role of scheduling in this result is highlighted by the contrasting survival outcomes seen between the two arms containing olaparib. The clinical results show that the introduction of a 48 h gap after commencing carboplatin-containing chemotherapy, but before starting olaparib (research arm), was integral to attaining the improved survival outcomes. In contrast, starting olaparib 48 h before carboplatin-containing chemotherapy (dropped arm) resulted in worse survival outcomes than for the chemotherapy alone (control arm). Since both schedules involved the same olaparib dose and duration, the order in which the agents were given seems the most likely explanation for the differential survival outcomes.

The majority of the cancer related events in these high-risk cancers would be expected in the first 36 months. Other gBRCAm and TNBC trials have reported at this timepoint[6,28,29]. The median follow-up in the trial population is 40 months, at which point only 1 out of 39 gBRCAm patients in the research arm had relapsed (local and distant relapse (bone metastases)). At 36 months there had been no deaths in the research arm.

The outcome differences between the two olaparib-containing arms reported here and the differences in results between gBRCAm and TNBC gBRCAwt[13] all support the likelihood that there is a biological rationale underlying these results. Acknowledging the limitations of the preclinical in vitro studies that addressed the effect of giving the olaparib before or after carboplatin, the data highlight that there can be differences in the effects of the two different combination schedules at the cellular level. The olaparib first schedule in the gBRCA1m TNBC cell line SUM149PT (akin to the dropped arm) reduced the proportion of cells in S-phase, where carboplatin will have its greatest impact in terms of DNA damage and replication stress induction, resulting in lower levels of observed DNA damage and cell kill relative to the carboplatin first gap schedule (akin to the research arm) (Fig. 6A–D). These laboratory results should be treated as hypothesis-generating, but they do highlight the importance of scheduling on DNA damage and cell viability, which in turn can impact anti-tumour efficacy. Translational science biomarkers for cell cycle status, DNA damage and cell death should be incorporated into future clinical studies for PARPi chemotherapy combinations.

Previous chemotherapy-PARPi concurrent schedules encountered dose-limiting bone marrow toxicity, which meant that the use of sub-maximally tolerated doses provided no clinical benefit versus platinum-based chemotherapy alone[22]. The PARTNER trial control and research arms resulted in similar treatment discontinuations due to toxicity and comparable QoL. The gap scheduling approach has the potential to be used for other PARPi and novel agent chemotherapy combinations and in other tumour types where gBRCAm populations are prevalent, such as in ovarian cancer.

Based on the OlympiA[9] study results, patients with residual disease post-neoadjuvant therapy are recommended to receive 12 months of olaparib. The research arm in the PARTNER trial could provide a regimen that reduces the exposure of younger breast cancer patients (median age 41.8 years) to longer-term treatment with PARPi that could interfere with the return of fertility and increase the potential for longer-term genotoxic effects. The PARTNER trial did not include immunotherapy[30] and therefore avoids the multiple short- and long-term toxicities associated with such treatments that are seen in early-stage breast cancer patients. Currently, there is insufficient published data to fully understand if, and to what extent, both pCR and survival rates are improved specifically in gBRCAm patients treated with chemotherapy and immunotherapy. For gBRCAm patients in the post-neoadjuvant setting, it is unclear which agents should be given, in which order, and for what specific additional benefit. Neoadjuvant olaparib does have an additional survival benefit for gBRCAm patients in the PARTNER trial research arm, irrespective of pCR. Where treatment schedules show similar survival rates, clinicians should use the least toxic and most cost-effective schedule.

The results of this study require that health services can test for gBRCAm, and potentially other pathogenic variants causing HRR deficiencies, such as *PALB2*, upfront within the necessary timeframe to allow prompt initiation of neoadjuvant therapies.

The primary outcome measure for this trial was pCR. The results show that the addition of olaparib does not improve pCR rates. The results also show that pCR status in the gBRCAm subtype does not accurately predict survival outcomes. In total, we found four other studies in the neoadjuvant setting that provided a pCR/non-pCR analysis for the gBRCAm subtype, and two of these reported a disconnect between pCR/non-pCR and survival outcomes[15,16]. A similar lack of association between neoadjuvant chemotherapy response scores and survival outcomes has been seen in gBRCAm ovarian cancer[31]. In contrast, two retrospective studies showed the opposite; namely, that in gBRCAm patients, non-pCR did predict for poorer survival outcomes[32,33]. A formal meta-analysis of gBRCAm cohorts within neoadjuvant trials (Supplementary Tables 11 and 12) investigating the association of pCR status with survival outcomes may help guide our understanding of the relationship between pCR and survival in the gBRCAm subtype. Many post-neoadjuvant studies allocate treatment based on non-pCR status. If non-pCR status is not a good surrogate for poor survival in gBRCAm patients, then other potential surrogates, such as ctDNA kinetics or clearance during treatment, may help select patients for adjuvant therapies.

Importantly, the patients in the research arm of PARTNER did not receive adjuvant PARPi, capecitabine or immunotherapy treatment. Therefore, additional adjuvant therapies do not explain the lack of association between pCR status and survival. A further explanation could be that cancer cells in the surgical specimen, although visible, had no potential for further growth or metastatic spread. In all breast cancer subtypes, we know that some patients do not relapse despite the presence of residual disease. The gap schedule research arm treatment may have a differential and larger effect on micrometastatic disease, thus reducing the development of established metastatic disease. The precise mechanism for this needs further investigation but may include the additional effects of PARPi on cellular processes for example olaparib induced increased sensitivity to anoikis[34–36] or olaparib induced senescence[37]. This, in conjunction with other factors such as increased S-phase fraction cell death, could have contributed to the improved survival outcomes.

Potential limitations of this study include early recruitment cessation at the pre-planned interim analysis time-point, leading to a smaller cohort for primary and secondary analyses. Across the two arms in the primary analysis, although most variables are well balanced, there is an imbalance in two variables (Supplementary Table 6). This imbalance is likely to be for two reasons: firstly, at recruitment, TNBC patients with unknown gBRCAm status were permitted to be randomised, but if later found to be gBRCAm they would contribute only to the gBRCAm analysis; and secondly, the analysis was completed at the planned interim analysis point rather than after recruitment of the full cohort.

Regardless of whether the imbalance favours the control arm or the research arm, the event rate is lower, and the EFS, OS and BCSS rates are all higher in the research arm. Therefore, these imbalances are unlikely to explain the improved survival outcomes. Inspecting the distribution of node positivity and when considering all three arms, there is a ranking in terms of survival. The selected research arm is superior to the control arm, and the control arm, in turn, is better than the dropped arm. In contrast, the node positivity has a different ordering (Supplementary Table 6). This result corroborates our findings that the observed imbalance in node positivity is not substantial and does not materially affect the results of this trial. Similarly, TILs score, which is imbalanced in favour of the control arm, also does not affect the results of the trial. These minor imbalances among the three arms do not explain the survival outcomes. Whilst acknowledging these potential limitations, the statistically significant differences in survival between the research arm, control arm and dropped arm clearly indicate that there is a potential underlying biological rationale for these different outcomes.

In conclusion, while the results from the PARTNER need confirming in a larger neoadjuvant trial, there is now the exciting possibility of delivering a highly effective, optimally scheduled combination regimen for the high-risk early-stage gBRCAm breast cancer population.

## Methods

This research complies with all relevant ethical regulations. Preclinical in vivo studies complied with all relevant ethical regulations for animal testing and research, followed AstraZeneca's global bioethics policy, and received ethical approval from the AstraZeneca ethical committee. The HBCx-17, HBCx-10 and HBCx-9 PDX model studies were carried out at XenTech, France, in accordance with French regulatory legislation concerning the protection of laboratory animals. The PARTNER trial protocol (NCT03150576 and EudraCT: 2015-002811-13) (Supplementary Note 4) was approved by Northwest–Haydock Research Ethics Committee (ref: 15/NW/0926) and the trial was performed in accordance with the Declaration of Helsinki and the European Clinical Trials Directives 2001/20/EC.

### Animal studies

**In vivo rat tolerability studies and peripheral blood analysis.** Male rats (strain RccHan:WIST age 12 weeks) were obtained from Harlan UK. The animals were allowed to acclimatise for at least 1 week and were multiple-housed up to 5 per cage. Water from the site drinking water supply and RM1 (E) SQC pelleted diet supplied by Special Diet Services Ltd., England, was freely available. Nesting material (Tapvei® aspen chips, Finland, Tapvei® small aspen bricks and sizzle nest) and polycarbonate tunnels (Datesand) were provided. Vehicle groups were administered with a slow intravenous (iv) bolus administration (via a tail vein over the course of a minute) of vehicle (iv formulation) and were dosed once daily by oral gavage with oral vehicle. Animals dosed with carboplatin were given a single slow bolus iv dose (via a tail vein over the course of a minute) of carboplatin (every 14 days in multiple cycle studies). Animals receiving olaparib were dosed once daily by oral gavage. Where olaparib and carboplatin were dosed on the same day, the olaparib (AZD2281) oral dose was given approximately 1 h before the carboplatin (iv) dose. The rats were euthanised by administration of halothane. Carboplatin was purchased from Tocris Bioscience and prepared in 0.9% NaCl. Olaparib was formulated in 10% v/v Dimethyl Sulfoxide (DMSO) + 90% v/v Hydroxy Propyl β Cyclodextrin (10% w/v in deionised water). For peripheral blood analyses, blood samples (0.4 ml in EDTA) were taken from the tail vein for assessment on days 3, 7 and 14. Haematology analysis was performed on an automated Siemens Advia 2120i analyser.

**Immunohistochemistry γH2AX analysis in rat bone marrow and tumour tissue.** Assessment of DNA damage using the biomarker of DNA double-strand breaks (DSBs) was carried out following carboplatin treatment in rats (40 mg/kg) or human xenografted mice (50 mg/kg). Rat femurs or human tumour samples implanted in mice were removed at necropsy and fixed in formalin in situ for 48 h prior to coring. Preserved tissue was processed into wax blocks, sectioned, and stained by immunohistochemistry for the presence of γH2AX (Ventana™, OminUltraMap HRP, Discovery XT Staining module, antibody γH2AX CST 25777 dilution 1:100). Sections were counterstained with haematoxylin. The γH2AX signal in individual samples was assessed by a pathologist under the microscope. The scoring system was 0–4 based on the proportion of positively stained cells and the staining pattern: 0 = no staining defined as less than 1–2 cells with >5 foci per cell within a section field; 1 = faint nuclear staining with a slight increase in the number of positive cells with distinct foci (>5 foci per cells); 2 = mild nuclear staining with an increase in the number of positive cells with distinct foci (>5 foci per cells); 3 = moderate nuclear staining with an increase in the number of positive cells with coalescing foci. The slides provided for analysis were scanned using the Aperio Scanscope, converted to TIFF images (×10 magnifications), and analysed on the KS400 image analyser.

### Flow cytometry analysis of bone marrow cells

Rat femurs were removed at necropsy, and both ends were trimmed. BM cells were immediately flushed out with 3 ml PBS containing 50%

foetal calf serum (FCS) using a 3-ml syringe and a microlance 3 needle (Becton, Dickinson and Company (BD)). Cell suspension was syringed and filtered through a 100 μm strainer, collected by centrifugation (300 g/7 min/4 °C), and washed once in HBSS containing 2% FCS and 10 mM HEPES (staining buffer). The total cell count of isolated cells was determined by an automated cell counter (Countess, Invitrogen). Cell concentration was adjusted to $1 \times 10^7$ cells/ml in staining buffer and processed for antibody staining. For the CD90.1 and Lineages cocktail, 1 ml of cell suspension was resuspended in 100 μl staining buffer containing anti-rat CD90.1-APC (dilution 1:100), CD6-FITC (1:100), CD3-FITC (1:100), CD11b-FITC (1:200), Granulocytes-FITC (1:200) from BD Pharmingen, and CD45RC-FITC (1:100) purchased from Bio-Rad (AbD Serotec). Cells were incubated with antibodies for 30 min at room temperature (RT) and washed twice in staining buffer. After the final centrifugation, 1 ml of staining buffer was added to CD90.1-Lineage-stained cells. Cells were incubated in the dark for 30 min prior to flow cytometry analysis. Data (at least 10,000 events) were acquired on the FACS Aria II (BD). Analysis was performed using FlowJo software. Data were expressed as a percentage of the total population. The total number of isolated BM cells was used to calculate the percentage of cells in the population of interest. For the ANOVA statistical analysis of $CD^{90+}$ cell levels, two-sided Student's $t$ comparisons were applied using pooled inter-individual variability with no adjustments for multiple comparisons and with response variables having been log transformed.

### In vivo PDX anti-tumour efficacy studies

For the three TNBC PDX models HBCx-10 (BRCA2m tumour), HBCx17 (BRCA2m tumour) and HBCx-9 (BRCA1 promoter methylated tumour), female Hsd-athymic Foxn1 nude mice were implanted subcutaneously with $3 \times 3 \times 3$ mm tumour and then assessed at day 28 of treatment for their response to olaparib alone (100 mg/kg daily for 28 d), with platinum alone (cisplatin on D0 at 6 mg/kg) and with a concurrent olaparib/platinum combination using the schedules and doses above for the single agents) (Supplementary Fig. 1). The olaparib response mirrored the platinum response, with HBCx-10 inducing tumour regression, HBCx17 an intermediate tumour stasis response, and HBCx-9 only minor tumour growth delay compared to the vehicle control. HBCx-17 was selected for the platinum-induced DNA damage (γH2AX IHC) and anti-tumour scheduling studies. For these studies, HBCx17 fragments (20mm³) were implanted subcutaneously in the interscapular area of 6–9 weeks old female nude mice (HSD:Athymic Nude-Foxn1nu, Harlan, France). Tumour growth was measured by bilateral calliper measurements, and the tumour volume was calculated using a standard formula. When the average tumour volume reached ~100 mm³, mice were randomised into treatment groups ($n = 10$ per group), based on the tumour volume, to achieve a comparable group mean across all groups prior to the start of treatments. Mice were dosed with Vehicle Control, carboplatin (50 mg/kg IP once on Day 1 of treatment), olaparib (100 mg/kg PO QDx28 from Day 3) or a combination therapy of carboplatin (50 mg/kg IP once on Day 1) plus olaparib (100 mg/kg PO QDx28 from Day 3). Tumour volume, body weight and clinical condition were measured up to 3 times a week and any effect on the tumour growth was calculated as percentage tumour growth inhibition relative to the Control Vehicle and/or individual monotherapy groups. The maximum tumour volume of ≤2000 mm³ was permitted before euthanisation, and this was not exceeded throughout this study. Statistical significance was evaluated using a one-tailed $t$-test.

### Cell culture of TNBC gBRCAm cell line SUM149PT

SUM149PT cells described in this study were obtained from Asterand Bioscience; authenticated by the AstraZeneca cell bank with short tandem repeat analysis using CellCheck (IDEXX Bioanalytics); and validated free of Mycoplasma contamination using the STAT-Myco assay (IDEXX Bioanalytics). Cells were grown in F12-Ham's media

supplemented with 5% FCS, 2 mM GlutaMAX, 500 ng/ml hydrocortisone and 0.01 mg/ml insulin.

### DNA damage and cell-cycle analysis of SUM149PT by immunofluorescence

Cells were seeded in 96-well plates (2000 cells/well) 72 h prior to treatment and were then treated with olaparib and carboplatin according to the schedules outlined in Fig. 6 and Extended Data Fig. 6. 5-Ethynyl-2'-deoxyuridine (EdU) was added for 30 min, cells were fixed for 15 min with 4 % formaldehyde (v/v) at RT, permeabilised with 0.2% Triton X-100 (v/v) in phosphate buffered saline (PBS) for 15 min and incubated in blocking solution (3% BSA (w/v), 10% FCS (v/v) in PBS) for 1 h at RT. EdU-positive cells were labelled following incubation with 2 mM CuSO4, 1 μM Alexa Fluor 647 azide and 10 mM sodium ascorbate in PBS for 1 h at RT. Following primary antibody incubation (overnight at 4 °C; anti-γH2AX, Merck clone JBW301; 1:5000 in 1% BSA (v/w), 0.1% Tween-20 PBS) and secondary antibody incubation for 1 h at RT (Alexa Fluor 488 goat anti-mouse; 1:2000 in 1% BSA (v/w), 0.1% Tween-20 PBS), DNA was counterstained with DAPI (Thermo Fisher 1 μg/ml in PBS) and acquisition and image analysis performed using a Yokogawa CV8000 automated confocal microscope, with a 20× air objective (20× LWD, NA = 0.45, LUCPLFLN20X) at a single z-plane. Images were analysed in Columbus (version 2.9.1, Perkin Elmer). Cell nuclei were identified, and intensity measurements were completed within that region only. For % EdU positive population a cut-off of background EdU signal was used at every experimental analysis. Dose–response curves were generated with GraphPad Prism Software (9.5.1).

### SUM149PT cell viability assays

SUM149PT cells were seeded in 384-well plates (Greiner) and incubated overnight under normal growth conditions (37 °C, 5% CO$_2$ and saturated humidity). Cells were dosed with olaparib and carboplatin using the Echo Liquid Handler (Beckman), either as single agents or in combination, for 6 days. To mimic differentiated drug combination schedules, growth media was replaced on the third and fifth treatment day, and cells were re-dosed. This allowed for switching from combination to single-agent exposure and for complete removal of both drugs. On day 6, cells were incubated with CellTiter-Glo reagent (Promega, 1:2 ratio) for 15 min in the dark at 37 °C. Consequently, luminescence was measured using a plate reader, and cell viability was calculated using Genedata software.

### Flow cytometry analysis

Cells were seeded in 6 cm dishes (NUNC) (150,000 cells/dish) 72 h prior to treatment and then treated with olaparib and carboplatin according to the schedules outlined in Fig. 6. On day four, cells were EdU treated for 30 min, fixed, permeabilised and EdU labelled according to the immunofluorescence protocol described above. Cells were resuspended in block buffer containing 1 μg/ml DAPI, and live, single, DAPI-positive cells (Extended Data Fig. 6) were analysed with the LSRFortessa benchtop flow cytometer (BD). Quantification of the number of cells in different stages of the cell cycle—gap 1 (G1), DNA synthesis (S) and the gap 2 and mitosis (G2/M) phases, was quantified using FlowJo Engine v5 (BD) software to establish the relative proportion of cells within these phases before and following treatment.

### Immuno-blotting and detection of specific proteins within the SUM149PT cells

Cells were lysed in RIPA buffer (Sigma-Aldrich) supplemented with protease inhibitors (Roche), phosphatase inhibitors (Sigma-Aldrich), and benzonase (Merck, catalogue No. 103773). Following protein concentration measurement, 50 μg of protein was mixed with NuPAGE LDS Sample Buffer (Thermo Fisher Scientific) and NuPAGE Sample

Reducing Agent (Thermo Fisher Scientific), and the samples were heated to 95 °C for 10 min. Whole cell lysates samples were separated on 4–12% Bis–Tris NuPAGE gels and analysed by standard immunoblotting using the following antibodies: γH2AX, 1:1000, (Milipore, 05-636), γH2AX, 1:1000 (Millipore, 07-627), phospho-RPA2, 1:1000 (S4/S8, Bethyl, A300-245), RPA2, 1:1000 (Bethyl, A300-244A), phospho-CDC2 T14, 1:1000 (Biolegend, 947402), CDC2, 1:1000 (Cell signalling Technologies (CST), 9116), PAR, 1:1000 (CST, 83732), Cleaved (Cl) PARP1, 1:1000 (CST, 9546), Cyclin A, 1:1000 (BD transduction, 611268), Vinculin, 1:5000 (Invitrogen, 700062)). Antibodies were detected with horse radish peroxidase (HRP)-conjugated secondary antibodies (anti-rabbit IgG, HRP, 1:1000 (CST, 7074), anti-mouse IgG HRP, 1:1000 (CST, 7076). Immunoblots are representative of experiments that were performed at least twice.

### Statistical analysis of SUM149PT cell line data

Data are presented as the mean ± SD. Statistical analyses were performed using GraphPad Prism 9.5.1 (GraphPad Software, Inc.).

### PARTNER Clinical Trial Design for the gBRCAm cohort

**Patients and treatments.** Patients aged between 16 and 70 years with histologically confirmed stage T1–4, N0–3 (tumour or axillary lymph node diameter ≥10 mm) invasive breast cancer, confirmed HER2 negative, and Eastern Cooperative Oncology Group performance status (ECOG PS) 0–1 were eligible. All patients had mandatory gBRCAm testing at trial entry. Patients identified as carriers of a deleterious germline (disease-causing) mutation in the *BRCA1* or *BRCA2* genes were included in this cohort independently of their tumour hormonal status. Other key inclusion criteria were patient fitness to receive the trial chemotherapy regimen and availability of slides and paraffin-embedded tissue blocks from the pre-treatment biopsy. Patients were excluded if they had a T0 tumour in the absence of axillary node ≥10 mm, apparent distant metastases, prior history of invasive breast cancer within the last 5 years or any previous chemotherapy or targeted agent used for the treatment of cancer in the last 5 years. All patients provided an initial written informed consent which covered pathological review of the local slides/biopsy tissue at the Cambridge Centre, with TILs assessment[38], and additional biomarkers (EGFR, CK5/6 and AR) to confirm basal-like TNBC. If the biomarkers confirmed this, the patient proceeded to full consent for the main study at the local centre. Patients who were confirmed ER-positive and HER2-negative but gBRCAm positive were allowed into the trial. This study was completed during the COVID-19 pandemic, and despite a large pause in recruitment, due to the dedication of our recruiting centres, recruitment to the gBRCAm interim analysis for futility was completed.

The trial design and dosing schedules are discussed in the main text. Patients were randomised using the minimisation method in a 1:1:1 ratio in Stage 1 and Stage 2 with a web-based central randomisation system.

Stratification factors included cancer type (TNBC/BRCA1/BRCA2/unknown), tumour size (≤50 mm/>50 mm), histopathological involvement of axillary nodes at diagnosis (yes/no), and TILs (<60%/≥ 60%). Granulocyte-colony stimulating factor was given as per local practice.

### Study procedures

Patients were clinically assessed prior to the beginning of every cycle until the end of treatment or disease progression. Breast surgery was performed after chemotherapy and was followed by radiotherapy as per local standard protocols. After surgery, patients were followed 6-monthly for 2 years and then annually for up to 10 years. Histopathology reports from primary surgery were centrally reviewed independently by two readers (E.P., plus one of H.M.E., L.D. and A.F.), who were blinded to the treatment arm and gBRCAm status, to determine the presence or absence of pCR. If there were any differences, a consensus for each patient was reached after discussion.

Adverse events (AEs) were reported for each cycle of protocol treatment using NCI CTCAE version 4.03. Quality of life (QoL) was optional and assessed using the EQ5D-L and FACT-B questionnaires prior to randomisation, following completion of four cycles, seven cycles, surgery and radiotherapy, and annually for 2 years from completion of surgery. Imaging was performed at baseline, completion of four cycles, and after full protocol treatment to assess the objective response rate (ORR) using RECIST V1.1.

## Statistical analysis

In this 3-stage phase II–III trial, stage 1 assessed the safety of the addition of olaparib to weekly paclitaxel and 3 weekly carboplatin chemotherapy. Stage 2 selected the "winner" from two research groups. Stage 3 assessed pCR at surgery after neoadjuvant treatment in all patients. The primary endpoint is a comparison of pCR between the research and control groups. Further details of the statistical analysis and IDMSC decision are provided in Supplementary Note 1. Secondary survival endpoints (Supplementary Note 2) were all calculated from the date of randomisation and included; (i) EFS[39]: local or distant recurrence, diagnosis of a second cancer, or death from any cause; (ii) relapse free survival (RFS)[40]: local or distant recurrence or death from any cause, excluding patients who relapsed before surgery; (iii) breast cancer specific survival (BCSS): death from breast cancer or death after breast cancer relapse; (iv) distant disease-free survival (DDFS)[40]: distant recurrence or death from any cause; (v) local recurrence-free survival (LRFS): local-recurrence or death from any cause; (vi) OS[40]: death from any cause; (vii) time to second cancer (TTSC): diagnosis of a second cancer. Other secondary endpoints were residual cancer burden (RCB); pCR in breast alone; radiological response; safety and QoL.

In this gBRCAm cohort, a total of 178 patients were needed to achieve a 90% power with a 5% significance level, assuming the pCR rate of 55% in the control group and 75% in the research group. Considering a non-compliance of 5%, it was planned to recruit a total of 188 gBRCAm patients between the control and the selected research group. The study design included an interim analysis for futility reported in the results of this paper. The trial was stopped at the interim analysis point on the advice of the IDMSC.

The treatment effect was estimated using a population defined based on the mITT principle, while the safety of the experimental treatment and all other analyses included patients who had at least one dose of trial treatment. The differences between binary outcomes were compared using the Chi-squared test, and the confidence interval of the differences was based on the score method[41]. Kaplan–Meier plots were generated for time-to-event outcomes, and groups were compared with the log-rank test. The subscales of EQ-5D-5L and FACT-B questionnaires were derived according to standard-scoring manuals. Analyses of changes from baseline over time and differences between the two treatment groups for subscales were carried out using repeated measures ANCOVA, adjusting for baseline level, time, treatment and interaction of time and treatment. It was assumed that the data were 'missing at random', but a sensitivity analysis for data 'missing not at random' was performed for the primary endpoint.

All statistical analyses were carried out in R (v4.1.0), and all P values are based on two-tailed tests.

## Reporting summary

Further information on research design is available in the Nature Portfolio Reporting Summary linked to this article.

## Data availability

Source data files have been provided with this article. The data regarding the baseline patient information, survival outcomes, the trial protocol and other detailed therapeutic information have been provided as Supplementary information and within the Article. De-identified data collected in the PARTNER study will be made available to researchers whose full proposal for their use of the data has been approved by the PARTNER trial management group and whose research includes a clear and comprehensive research plan with statistical considerations adequately completed. The data required for the approved, specified purposes will be provided after completion of a data sharing agreement. Data sharing agreements will be set up by the trial management groups and will include clear instructions on publication, reporting and usage policy. A minimum dataset of anonymized data will be made available after full publication of the trial and related work. Requests for data should be addressed to ja344@cam.ac.uk. Source data are provided with this paper.

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

## Acknowledgements

Our thanks go first and foremost to the patients, the families, and friends who supported them for participating in this trial. This trial was sponsored by Cambridge University Hospitals NHS Foundation Trust and the University of Cambridge. It was funded by a project grant from AstraZeneca, which also supplied olaparib. Cancer Research UK provided peer review and endorsement for the study and funded the sample collections for the translational studies, which will be reported separately. The sponsors and the funders had no role in data collection or analysis. Once the trial group had interpreted the data, AstraZeneca scientists were then approached to understand if their preclinical data could provide an explanation for differences in outcomes across the trial arms and to provide preclinical data for the schedule concept. We would like to thank those current and past AstraZeneca scientists, who are not co-authors but who nevertheless contributed to the preclinical data being generated, namely Catherine Wilkinson, Richard Knights, Pete Newham, Alan Lau, James Yates, Rajesh Odedra and Elaine Cadogan. We thank our ethics committee, our independent data and safety monitoring committee, and the trial management and steering groups for their diligence and thoughtfulness in their advisory roles. We thank Ms Ingrid Cizaite for proofreading the paper. In addition, we thank: the NIHR Cambridge Biomedical Research Centre and the Cancer Research UK Cambridge Centre Cambridge for their support for staff and infrastructure costs; Cancer Molecular Diagnostics Laboratory and the Precision Breast Cancer Institute Team for their support for sample collection; the Cambridge Tissue Bank (NIHR203312) for sample assessment and diagnostics; the Cambridge Clinical Trials Centre—Cancer Theme for their core staff support; the clinical trials support staff at all participating sites, and Addenbrooke's Charitable Trust for funding the Chief Investigator (2015-2018). The PARTNER trial was supported by Cancer Research UK [CRUKE/14/048] and AstraZeneca [1994-A093777].

## Author contributions

J.E.A.: Chief Investigator, overall oversight, design, management, data interpretation and paper writing for the PARTNER trial and site PI for coordinating central site; L.O.O.C.: Preclinical experimental design, data acquisition, data analysis, paper writing; L.G.: Senior trials coordinator dealing with day to day management of the trial and all the trial team; K.P.: Clinical fellow contributing to paper writing, data interpretation, trial safety management and protocol development; A.D.: Main statistician dealing data analysis of the PARTNER trial; N.D.: Senior statistician over-seeing data analysis of the PARTNER trial; C.H.: Senior data manager overseeing data collection team; L.D.: Clinical Fellow, contributing to trial safety management and pathology report assessment; R.L.: Clinical Fellow, contributing to trial safety management and protocol development; A.F.: Clinical Fellow, contributing to trial safety management and pathology report assessment; A.R.: Clinical Trials Practitioner and contributed to Case Report Development; J.W.: Senior Clinical Trials

Practitioner and contributed to Case Report Development and sample management; A.C.: Senior Pharmacist contributed to trial development; W.Q.: Statistical design and paper writing; J.B.: Clinical fellow and trial design; R.H.: Lead Programmer for development of Case Report Form and data collection proformas; A.L.V.: Senior trial co-ordination and trial design; C.S.: Preclinical experimental design, data acquisition and analysis; J. Ba: Preclinical experimental design, data acquisition and analysis; D.P.: Preclinical experimental design, data acquisition, data analysis, paper writing; P.W.: Preclinical experimental design, data acquisition, data interpretation, paper writing; G.Z.V.: Preclinical Data acquisition. S.G.: Paper writing; M.J.O.C.: Preclinical concept, experimental design, supervision, data analysis and interpretation, paper writing and paper revision; Site PIs (recruitment and consent of patients): S.C.; M.C.; E.S.: A.C.; M.P.; S.S.; S.R.; A.B.; J. Bra; E.S.; L.S.; C.P.; M.M.; M.C.; M. Mu; R.R.; N.L. Site PIs with Trial Management Group responsibilities: J.N.; K.Mc A.; A.A. (co-chair of TMG); E.C. (co-chair of TMG); P.S.: trial pathologist; E.M.: trial development and olaparib drug access; editing; M.T.: Medical Genetics Lead; E.P.: Senior trial pathologist; H.M.E.: Design, management, data interpretation and paper writing for the PARTNER trial. A full list of the PARTNER trial group is available in Supplementary Note 5.

## Competing interests

The funders of the research grants and honoraria had no role in the study design, data collection, analysis, interpretation, or writing of the report. The authors declare the existence of the following competing interests: J.E.A. reports honoraria, conference attendance, travel support, and a grant from AstraZeneca; and honoraria from Esai and Pfizer for lectures. L.O.O.C. reports employment and shares in AstraZeneca. J.B. reports employment and stock in AstraZeneca. C.S. reports former employment and stock in AstraZeneca. J. Ba reports former employment and stock in AstraZeneca. M.M. reports shares in AstraZeneca. D.P. reports employment and shares in AstraZeneca. G.Z.V. reports employment and shares in AstraZeneca. P.W. reports employment and shares in AstraZeneca. M.B.M. reports advisory board membership in Roche, Pfizer, MSD, Daiichi-Sankyo, Gilead, AstraZeneca, Novartis, Menarini group, Genomic Health (Precision Medicine) & Seagen; speaker honoraria from Roche, BMS, Seagen, Pfizer, Daiichi-Sankyo, AstraZeneca, Lilly, MSD, Genomic Health (Precision Medicine), Eisai & Novartis; and meeting expenses from Roche, Eli Lilly, Novartis and MSD. R.R.R. reports honoraria from Daiichi-Sankyo, AstraZeneca, Novartis, Pfizer; membership in advisory boards for Daiichi-Sankyo, Eli Lilly, Pfizer, AstraZeneca; and travel/conference attendance for BMS, Pfizer, Roche. PCS reports that their partner is employed by AstraZeneca. N.C.L. reports shares in AstraZeneca. A.C.A. reports research funding paid to the Institution from AstraZeneca; conference fees and travel expenses from Roche and Novartis; conference fees from M.S.D.; membership on Roche and AstraZeneca advisory boards; and a grant for an educational project from Gilead. ERC reports honoraria from AstraZeneca, Eli Lilly, Novartis, Pfizer, Roche; membership in advisory boards for AstraZeneca, Eli Lilly, Pfizer, Menarini Stemline UK, Novartis; consultancy for Pfizer, conference fees/travel/accommodation from Roche, Novartis; an educational grant from Daiichi-Sankyo; and research funding and support from SECA, AstraZeneca. S.G. reports employment and stock in AstraZeneca. EP reports honoraria from Roche, Novartis, and AstraZeneca. M.J.O.C. reports employment and shares in AstraZeneca.

## Additional information

Jean E. Abraham [1,2] ✉, Lenka Oplustil O'Connor [3], Louise Grybowicz[4], Karen Pinilla Alba [1,2], Alimu Dayimu[5], Nikolaos Demiris[6], Caron Harvey[4], Lynsey M. Drewett[7], Rebecca Lucey [1,2], Alexander Fulton[1,2], Anne N. Roberts[4], Joanna R. Worley[1,2], Ms Anita Chhabra [8], Wendi Qian [9], Jessica Brown[3], Richard Hardy[5], Anne-Laure Vallier[4], Steve Chan[10,11], Maria Esther Una Cidon[12], Elizabeth Sherwin[13], Amitabha Chakrabarti[14], Claire Sadler[15], Jen Barnes[3], Mojca Persic[16], Sarah Smith[17], Sanjay Raj[18,19], Annabel Borley[20], Jeremy P. Braybrooke[21], Emma Staples[22], Lucy C. Scott[23], Cheryl A. Palmer[24], Margaret Moody[25], Mark J. Churn[26], Domenic Pilger [3], Guido Zagnoli-Vieira[3], Paul W. G. Wijnhoven[3], Mukesh B. Mukesh[27], Rebecca R. Roylance [28], Philip C. Schouten[29], Nicola C. Levitt[30], Karen McAdam[31], Anne C. Armstrong [32], Ellen R. Copson[33], Emma McMurtry[34], Susan Galbraith[3], Marc Tischkowitz [35], Elena Provenzano[29], Mark J. O'Connor [3], Helena M. Earl[36] & PARTNER Trial Group

[1]Precision Breast Cancer Institute, Department of Oncology, University of Cambridge, Cambridge, United Kingdom. [2]Cancer Research UK Cambridge Centre, University of Cambridge, Cambridge, United Kingdom. [3]AstraZeneca, Cambridge, United Kingdom. [4]Cambridge Cancer Trials Centre, Cambridge University Hospitals NHS Foundation Trust, Cambridge, United Kingdom. [5]Cambridge Clinical Trials Centre, Cancer Theme, University of Cambridge, Cambridge, United Kingdom. [6]Department of Statistics, Athens University of Economics and Business, Athens, Greece. [7]Royal Devon University Healthcare

NHS Foundation Trust, Exeter, United Kingdom. [8]Cambridge University Hospitals NHS Foundation Trust, Cambridge, United Kingdom. [9]Cambridge Clinical Trials Unit, Cambridge University Hospitals NHS Foundation Trust, Cambridge, United Kingdom. [10]The City Hospital, Nottingham University Hospitals NHS Trust, Nottingham, United Kingdom. [11]Nottingham Breast Cancer Research Centre, University of Nottingham, Nottingham, United Kingdom. [12]Royal Bournemouth General Hospital, University Hospitals Dorset NHS Foundation Trust, Bournemouth, United Kingdom. [13]Ipswich Hospital, East Suffolk and North Essex NHS Foundation Trust, Ipswich, United Kingdom. [14]University Hospitals Dorset NHS Foundation Trust, Poole, United Kingdom. [15]Apconix Ltd, Alderley Edge, Cheshire, United Kingdom. [16]University Hospital of Derby and Burton, Derby, United Kingdom. [17]Bedford Hospital, Bedfordshire Hospitals NHS Foundation Trust, Bedford, United Kingdom. [18]University Hospital Southampton NHS Foundation Trust, Southampton, United Kingdom. [19]Hampshire Hospitals NHS Foundation Trust, Hampshire, United Kingdom. [20]Velindre Cancer Centre, Cardiff, United Kingdom. [21]University Hospitals Bristol and Weston NHS Foundation Trust, Bristol, United Kingdom. [22]Queens Hospital, Barking, Havering and Redbridge University Hospitals NHS Trust, Romford, United Kingdom. [23]Beatson West Of Scotland Cancer Centre, Glasgow, Scotland, United Kingdom. [24]Hinchingbrooke Hospital, North West Anglia NHS Foundation Trust, Huntingdon, United Kingdom. [25]Macmillan Unit, West Suffolk Hospital NHS Foundation Trust, Bury St Edmunds, United Kingdom. [26]Worcestershire Acute Hospitals NHS Trust, Worcester, United Kingdom. [27]Colchester General Hospital, East Suffolk & North Essex NHS Trust, Colchester, United Kingdom. [28]University College London Hospitals NHS Foundation Trust, London, United Kingdom. [29]Department of Histopathology, Addenbrooke's Hospital, Cambridge University Hospitals NHS Foundation Trust, Cambridge, United Kingdom. [30]Oxford University Hospital NHS Foundation Trust, Oxford, United Kingdom. [31]Peterborough City Hospital, North West Anglia NHS Foundation Trust, Peterborough, United Kingdom. [32]The Christie NHS Foundation Trust, Manchester, United Kingdom. [33]Cancer Sciences Academic Unit, University of Southampton, Southampton, United Kingdom. [34]EMC2 Clinical Consultancy Ltd, Sale, United Kingdom. [35]Department of Genomic Medicine, National Institute for Health Research Cambridge Biomedical Research Centre, University of Cambridge, Cambridge, UK. [36]Department of Oncology, University of Cambridge, Cambridge, United Kingdom. A list of members and their affiliations appears in the Supplementary Information. ✉e-mail: ja344@cam.ac.uk

