## [Transparent Peer Review file · Nature Communications]

Neoadjuvant PARP inhibitor scheduling in BRCA1 and BRCA2 related breast cancer: PARTNER - a randomized phase II/III trial

Corresponding Author: Professor Jean Abraham

Version 0:

Reviewer comments:

Reviewer #1

(Remarks to the Author)

The authors responded well to most of the concerns of the reviewers. However, two substantial ones remain: the full rationale for early termination of the study, and the exclusion of two control group patients from the primary analysis, without at least presentation of such results as a sensitivity analysis.

1. I do not think that the answer given in the manuscript for the early termination of the study is very good. The statement is: "Results from this analysis were reviewed by the IDMSC, which concluded that, while the pre-determined statistical criteria for futility (conditional power < 15%) were not met, it was clear that the primary endpoint (pCR) was unlikely to be achieved". The bar for futility was set at 15%, which is a reasonable bar, but then the study was stopped with a conditional power of 28.6%, nearly double the "pre-determined" futility bar. Such a probability does not make it "clear" to many researchers including myself that the trial might ultimately have succeeded. In the reviewer response, the authors noted that the posterior predictive probability was < 1%. How could this be if the conditional power is 28.6%? Another reason given in the response to the reviewers was that "the current pCR rate in the research arm was numerically lower than the control arm." The conditional power takes the current data into account, so there isn't a particular need for this to be used as a separate criterion. It is unclear, whether the IDMSC had authority to close the study or only advisory power. A stronger rationale and ownership of who had decision-making authority for closing the trial should be given in the manuscript. As stated, it appears that the study was abandoned rather than closed for a compelling reason.

2. Two reviewers independently pointed out that the most standard approach, that of a modified intention-to-treat analysis, for a randomized clinical trial, which is to include all randomized and treated patients. Thus, the sample size for the control group would be 45, not 43 as given. I believe that, at minimum, they should also report the results under that scenario in the manuscript, perhaps calling it a sensitivity analysis, and stating that the results do not substantially differ.

Reviewer #2

(Remarks to the Author)

Given that this is the second rebuttal, I am satisfied with the answers. There are some issues that could still be debated, but overall, the interesting findings outweigh the remaining shortcomings.

Reviewer #3

(Remarks to the Author)

I had the pleasure of re-reviewing the article on the PARTNER trial in the gBRCA mutated population.

The authors responded very forcefully to most of my previous comments.

At this stage, my comments only concern the form of the discussion. The authors' confidence and optimism could be praised, but I think it's important to stay as close to reality as possible: this negative trial is interesting, but it has no vocation to

change contemporary practices, and the proposed treatment regimen cannot reasonably be considered as a treatment option in clinics.

I therefore apologize for urging the authors to:

- Shorten the discussion, which seems to me too long and very speculative (e.g. trying to explain why cancer cells may be "doomed" although still alive is)
- Acknowledge that currently available real-life results with the KN522 regimen report pCR rates in gBRCA patients that are numerically (far) superior to the pCR reported in their trial, e.g. 82% (10.1016/j.annonc.2024.10.023) and 80% (10.1016/j.esmorw.2024.100061). Even if these recent series are still limited, their number will increase - I think the authors should rather acknowledge this signal than deny it.
- Remove all sentences relating to a potential use or implementation of the results in the clinics (e.g., line 307 and beyond, where it is suggested that the results can be used by clinicians, and that the results of this study "require" health services to test for gBRCA in real time, etc).

GENERAL COMMENT TO ALL REVIEWERS: We are indebted to the reviewers for the time and effort spent on reviewing our manuscript. Their comments have been helpful in improving the clarity of the manuscript's messages.

RESPONSES TO INDIVIDUAL REVIEWERS' COMMENTS

Reviewer #1 (Remarks to the Author):

The authors responded well to most of the concerns of the reviewers. However, two substantial ones remain: the full rationale for early termination of the study, and the exclusion of two control group patients from the primary analysis, without at least presentation of such results as a sensitivity analysis.

1. I do not think that the answer given in the manuscript for the early termination of the study is very good. The statement is: "Results from this analysis were reviewed by the IDMSC, which concluded that, while the pre-determined statistical criteria for futility (conditional power < 15%) were not met, it was clear that the primary endpoint (pCR) was unlikely to be achieved". The bar for futility was set at 15%, which is a reasonable bar, but then the study was stopped with a conditional power of 28.6%, nearly double the "pre-determined" futility bar. Such a probability does not make it "clear" to many researchers including myself that the trial might ultimately have succeeded. In the reviewer response, the authors noted that the posterior predictive probability was < 1%. How could this be if the conditional power is 28.6%? Another reason given in the response to the reviewers was that "the current pCR rate in the research arm was numerically lower than the control arm." The conditional power takes the current data into account, so there isn't a particular need for this to be used as a separate criterion. It is unclear, whether the IDMSC had authority to close the study or only advisory power. A stronger rationale and ownership of who had decision-making authority for closing the trial should be given in the manuscript. As stated, it appears that the study was abandoned rather than closed for a compelling reason.

Response: The study was not abandoned. Our IDSMC was very experienced, and their capacity was advisory. They advised the closure of the trial. The Trial Steering Committee (TSC), which does have the rights and responsibilities of being able to stop the trial, followed their advice.

The advice of the IDSMC was based on a number of considerations, in addition to the main criterion which was the conditional power. Please note that the posterior predictive probability of achieving the trial target is a very different measure to the conditional power. The latter, in its default definition, is essentially a weighted average between the observed data and the assumption of the alternative hypothesis (here a 20% improvement over the control arm) and, as such, it is generally less likely to stop a trial as it is affected by the alternative hypothesis. There are additional definitions of the conditional power, including based on the null hypothesis, and those were also presented to the IDSMC. The posterior predictive probability is based upon the current data (and a prior, essentially "worthy" of a single datum) and it calculates the probability

that the trial will be successful given the current observations. Therefore, the two notions are very different and perfectly compatible.

In summary, although the predetermined futility criterion (conditional power < 15%) was not met, the IDSMC members considered that, on balance, this study appeared unlikely to meet its primary endpoint and therefore they recommended it was stopped, giving patients the opportunity to undertake alternative treatment options.

During their discussions they took into account practical considerations regarding the landscape of treatment for gBRCA in the UK at that time. All recruiting sites were based in the UK. The approval for adjuvant olaparib in the UK in April 2023, to be given to post-neoadjuvant patients with residual disease, meant that if we had continued to recruit, our patients with residual disease after neoadjuvant therapy, would have received adjuvant olaparib for 12 months, and therefore there was a probability that the secondary endpoints would have been affected.

Therefore, taking into account data from the trial as seen by the IDMSC and the changes in standard of care, weighing all the evidence, they decided to recommend to the TSC that the trial should stop. The TSC and the trial management group acted on this advice.

For clarity we have added the following sentences:

“The IDMSC in their advisory role.....”

“This advice was considered by the trial steering committee and trial management group, who then stopped the trial.”

2. Two reviewers independently pointed out that the most standard approach, that of a modified intention-to-treat analysis, for a randomized clinical trial, which is to include all randomized and treated patients. Thus, the sample size for the control group would be 45, not 43 as given. I believe that, at minimum, they should also report the results under that scenario in the manuscript, perhaps calling it a sensitivity analysis, and stating that the results do not substantially differ.

Response: We showed in the sensitivity analyses that the outcome of these two patients do not affect the results of the trial's primary outcome. This is a statistically robust conclusion, and it is extensively discussed in the response to points 4 and 5 of Reviewer 1. This discussion including our response is now copied below for clarity:

Reviewer 1 Point 4: (p 7) Two of 45 patients were not included in the evaluation of the control arm. An intention-to-treat paradigm would use the result from the patient whose surgery came after the “data cutoff date”. For the remaining patient with unavailable tissue, the results could be given assuming the patient had a pCR or did not.

Response: One patient had surgery, but the tissue was not available by the data cutoff date. The other patient had no surgery confirmed by the site. One could assume that the missing outcome may be pCR or non-pCR with the risk of biasing the data.

We have performed complete case analysis as our baseline approach, and this method is unbiased when the data are missing at random. We have also performed a sensitivity analysis assuming data missing not at random with delta-adjusted pattern mixture model for completeness.

One could also assume that the missing observations are non-pCR whence the pCR rate would be 66.7% (30/45) and 64.1% (25/39) in the control and research arm, respectively. The difference in pCR rate between two arms is -2.6 (95% CI -22.9 to 17.6, p-value = 0.805). **Thus, the conclusion of the study would remain unaffected.**

Reviewer 1 Point 5: The Extended Data Table 4, with logistic regression analysis using a “delta-adjusted pattern mixture model” is both very difficult to interpret and not needed considering the simplicity of the imputation needed.

Response: Note that this discussion revolves around the two participants who had missing pCR outcome data. The statistical analysis plan was finalised prior to receiving the primary outcome data, thereby ensuring that the analysis was not subsequently modified by the data. The delta-adjusted pattern mixture model was selected, as a very commonly used sensitivity analysis to a potentially “missing not at random mechanism”. This is especially true in clinical trials because they allow transparent and clinically interpretable assumptions about the missing data distribution (Lu, K. . (2013). An analytic method for the placebo-based pattern-mixture model. *Statistics in Medicine*).

We have provided explanations in the footnote under the Extended Data Table 4.

Beyond the methodological discussion, note that in practice any adopted assumption about the outcomes of those 2 individuals does not affect the results of the paper. We are happy to be led by the editorial team with respect to removing the Extended Data Table 4.

Note that these two patients with missing pCR status were included in all the survival comparisons since those outcomes were available.

Reviewer #2 (Remarks to the Author)

Given that this is the second rebuttal, I am satisfied with the answers. There are some issues that could still be debated, but overall, the interesting findings outweigh the remaining shortcomings.

Response: We are indebted to this reviewer and all the others for the time and effort spent on reviewing our manuscript. We are truly grateful for their thoughts and comments.

Reviewer #3 (Remarks to the Author)

I had the pleasure of re-reviewing the article on the PARTNER trial in the gBRCA mutated population.

The authors responded very forcefully to most of my previous comments.

At this stage, my comments only concern the form of the discussion. The authors' confidence and optimism could be praised, but I think it's important to stay as close to reality as possible: this negative trial is interesting, but it has no vocation to change contemporary practices, and the proposed treatment regimen cannot reasonably be considered as a treatment option in clinics.

I therefore apologize for urging the authors to:

- Shorten the discussion, which seems to me too long and very speculative (e.g. trying to explain why cancer cells may be "doomed" although still alive is)

Response: Many post neoadjuvant trials now randomise patients to treatments based on the pCR status and their residual disease status (presence/absence). In the wild-type TNBC setting this may be perfectly sensible. However, in a setting where several trials (including this one) have shown a disconnect between pCR status and EFS/OS for gBRCA breast cancers it would not be an appropriate post-neoadjuvant strategy to allocate/randomise treatments based on pCR status alone.

It has long been known in all sub-types of breast cancer that some patients do not relapse despite having residual disease present in the surgical specimen. It is a valid biological question in this setting to ask why some patients, despite having residual disease, do not go on to get metastatic disease. If we can understand why the cancer cells in these patients have a disabled metastatic potential, we could then understand how to achieve the same biological status in other patients. Rather than being speculative, we are stating a fact and urging the scientific community to try and understand why this phenomenon occurs. We have nonetheless removed the term "doomed" but have retained the discussion around the biological concepts mentioned above.

- Acknowledge that currently available real-life results with the KN522 regimen report pCR rates in gBRCA patients that are numerically (far) superior to the pCR reported in their trial, e.g. 82% (10.1016/j.annonc.2024.10.023) and 80% (10.1016/j.esmorw.2024.100061). Even if these recent series are still limited, their number will increase - I think the authors should rather acknowledge this signal than deny it.

Response: The reviewer cites an ESMO abstract (full paper not published) and a paper in which the main message is the increased toxicity seen by the use of the KN522 regimen. The paper contains retrospective observational cohort analysis. When

considering the evidence from these two small observational retrospective studies it has to be acknowledged that the results of these studies because they are not randomised may be affected by patient selection and other confounders. In contrast the results in our study which come from a randomised controlled trial are not affected by such confounding factors. Whilst real-world data is important, the issue of confounders and even smaller sample size limits the value and therefore the interpretation of these studies, particularly where a full peer-reviewed publication is unavailable.

We fully acknowledge that our trial result needs validating in a larger and adequately powered study. One of the key messages from this trial is that we have demonstrated, like others, that pCR may not be a good surrogate for survival for gBRCA patients. Therefore, the discussion about how good the pCR rate is in any study needs to be tempered by the survival data. No survival data has been provided for either of the smaller retrospective observational studies mentioned and there is a potentially incorrect assumption that pCR will be associated to survival.

The disconnect we observed in PARTNER between pCR and survival outcomes is also seen in the biomarker data from the Keynote 522 trial presented by Joyce O'Shaughnessy at SABCS 2024*. Below is the data on Homologous Recombination Deficit (HRD). HRD is the closest surrogate available for gBRCA patients from the data released by Keynote 522:

* O'Shaughnessy J. Exploratory biomarker analysis of the phase 3 KEYNOTE-522 study of neoadjuvant pembrolizumab or placebo plus chemotherapy followed by adjuvant pembrolizumab or placebo for early-stage TNBC. Presented at the 2024 San Antonio Breast Cancer Symposium; December 10-13, 2024; San Antonio, TX. LB1-07.

[redacted]

You will note that the pCR benefit does not equate to EFS benefit.

Of the HRD positive cohort in KN522, 17% were gBRCA positive.

- Remove all sentences relating to a potential use or implementation of the results in the clinics (e.g., line 307 and beyond, where it is suggested that the results can be used by clinicians, and that the results of this study "require" health services to test for gBRCA in real time, etc).

Response: In addition to the HRD data mentioned previously, a further plot from the SABCS 2024 presentation shows:

In the HRD positive subgroup (a superset of gBRCA), there is a significant difference in terms of EFS but not in terms of pCR. So again, the connection between pCR and EFS is largely absent. The EFS rates at 36 months on the IO containing arm above readout at EFS~85-86%.

In comparison, in the successful research arm in gBRCA PARTNER trial, the 36-month EFS was 100%. We therefore maintain that these results, if confirmed in a larger trial, could change practice. We have also discussed these results with funders and patient advocates all of whom agree that a confirmatory study would be of interest and the results if validated have the potential to change practice.

In either case, it will be impossible to personalise treatments for gBRCA patients without upfront rapid gBRCA testing in the neoadjuvant setting. We believe it is important that in countries like the UK with a public health service the importance of rapid testing should be highlighted.

For clarity we have added the following to the text:

“Currently there is insufficient published data to understand if, and to what extent, both pCR and survival rates are improved specifically in gBRCAm patients treated with chemotherapy and immunotherapy. Neoadjuvant olaparib does have an additional survival benefit for gBRCAm patients in the PARTNER trial research arm, irrespective of pCR. Where treatment schedules show similar survival rates, clinicians should use the least toxic and most cost-effective schedule.”

“Many post-neoadjuvant studies allocate treatment based on non-pCR status.”

“It is well established in all breast cancer sub-types that some patients do not relapse despite the presence of residual disease.”

“The precise mechanism for this needs further investigation but may include the additional effects of PARPi on cellular processes for example olaparib induced increased sensitivity to anoikis^{1,2,3} or olaparib induced senescence⁴”

We have removed the following to shorten the discussion:

“There is no definitive evidence to show that immunotherapy has a specific additional benefit to chemotherapy in gBRCAm patients, whereas it is clear that neoadjuvant olaparib does have an additional survival benefit as seen in the PARTNER trial research arm.”

“Currently, clinicians are wrestling with challenging decisions in gBRCAm patients regarding whether to give adjuvant immunotherapy, adjuvant olaparib, or both. These results potentially simplify that decision.”

“To our knowledge, the PARTNER trial is the largest prospective neoadjuvant trial of PARPi in gBRCAm patients (Supplementary Table 11).”

“A better understanding of this relationship is required as future studies may select adjuvant treatments based on pCR/non-pCR status.”

“Careful histopathological review of the ‘residual cancer cells’ for the non-pCR gBRCAm cases has not revealed any obvious distinguishing features macroscopically that would help us understand why, for the research arm, in the patients with residual disease there was only one relapse and no deaths.”

“We hypothesise that the research arm treatment may affect potential micrometastatic tumour cells at the point of anoikis¹. Anoikis, apoptosis caused by inadequate cell-matrix interactions, occurs when the tumour cell leaves the primary tumour. The research arm treatment could increase sensitivity to anoikis and tumour cell death, thus preventing the transition to metastatic disease. This manifests clinically with the improved rates of EFS and OS. Although the precise mechanism of this effect requires further investigation, one hypothesis is that PARP inhibition can increase anchorage independent cell death and can induce anoikis and reduce resistance^{2,3}.”

References

1. Yao, D., Dai, C. & Peng, S. Mechanism of the Mesenchymal–Epithelial Transition and Its Relationship with Metastatic Tumor Formation. *Mol. Cancer Res.* **9**, 1608–1620 (2011).
2. Choi, E.-B. *et al.* PARP1 enhances lung adenocarcinoma metastasis by novel mechanisms independent of DNA repair. *Oncogene* **35**, 4569–4579 (2016).
3. Prasad, C. B. *et al.* Olaparib modulates DNA repair efficiency, sensitizes cervical cancer cells to cisplatin and exhibits anti-metastatic property. *Sci. Rep.* **7**, 12876 (2017).
4. Wang, Z., Gao, J., Zhou, J., Liu, H. & Xu, C. Olaparib induced senescence under P16

or P53 dependent manner in ovarian cancer. *J. Gynecol. Oncol.* **30**, (2019).